# Genetic Diversity of Canine *Babesia* Species Prevalent in Pet Dogs of Punjab, Pakistan

**DOI:** 10.3390/ani9070439

**Published:** 2019-07-13

**Authors:** Muhammad Tayyub, Kamran Ashraf, Muhammad Lateef, Aftab Ahmad Anjum, Muhammad Asad Ali, Nisar Ahmad, Muhammad Nawaz, Muhammad Mudasser Nazir

**Affiliations:** 1Department of Parasitology, University of Veterinary and Animal Sciences, Lahore 54000, Pakistan; 2Department of Microbiology, University of Veterinary and Animal Sciences, Lahore 54000, Pakistan; 3Department of Pathobiology, Bahauddin Zakariya University, Multan 66000, Pakistan

**Keywords:** babesiosis, pet dogs, Lahore, Narowal, Pakistan, *Babesia canis*, *Babesia gibsoni*

## Abstract

**Simple Summary:**

Babesiosis is caused by the intra-erythrocytic *Babesia* species in dogs. Different species are reported worldwide. The present study was conducted on pet dogs of two districts of Punjab, Pakistan, including Lahore and Narowal. Conventional microscopic and molecular tests, including conventional and nested PCR, were used for identification of diseased dogs and prevalent species. About 42% of dogs were tested positive for babesiosis and only two species including *Babesia canis* and *gibsoni* were prevalent.

**Abstract:**

Canine babesiosis is a serious threat to dogs’ health worldwide, caused by the intra-erythrocytic *Babesia* species. The present study was carried out in pet dogs presented at three clinics of Lahore and one of Narowal in Punjab, Pakistan. Two hundred blood samples (50 from each clinic) were collected and screened by microscopy for *Babesia* spp. Out of 200 samples, 84 (42%) were found to be positive for babesiosis. The highest number of positive cases (50%) was recorded in dogs at Narowal clinic. Non-significant variation (*p* > 0.05) was observed in the prevalence of babesiosis in dogs in relation to sex and age. Positive samples were further confirmed by a polymerase chain reaction using 18S-rRNA genus-specific and species-specific primers. Amplicons were further analyzed by nucleotide sequencing for genetic diversity. *Babesia canis* and *gibsoni* were confirmed by genome sequencing in all diseased dogs. These isolates closely resembled each other, but differed from previous reported strains. In conclusion, pet dogs suffering from babesiosis were infected with *B. canis* and *gibsoni*, while in other countries, other *Babesia* species are also prevalent.

## 1. Introduction

Babesiosis is a protozoan disease which is caused by intra-erythrocytic protozoa. In the present era, it has been distributed all over the world. Various species of *Babesia* infect vertebrates; however, the major etiological agents of canine babesiosis are *Babesia canis* and *gibsoni.* Canine babesiosis can be characterized by progressive anemia, fever, thrombocytopenia, and marked splenomegaly [1]. In severe cases, babesiosis can result in death. *Babesia canis* has a large piroplasm. Its length is generally measured as 4–5 µm. However, *B. gibsoni* has a small piroplasm and occurs in North America, Northern Africa, Eastern Africa, Europe, and Asia [2]. *Babesia gibsoni* measures to 3 µm in length. Therefore, *B. canis* is taken for a large piroplasm and *B. gibsoni* for a small one [3]. Babesiosis can be diagnosed by several methods and techniques in medical science. However, the most common and cheap method is microscopic examination of blood smear from an infected individual. Its accuracy is limited because sometimes, it does not provide accurate results, but it is considered a feasible and the simplest method. Several other methods have been reported for detection, included enzyme-linked immune sorbent assay (ELISA). Enzyme-linked immune sorbent assay is a reliable and valuable tool for Babesiosis detection but it has also some limitation [4]. In recent years, molecular methods such as polymerase chain reaction (PCR) gained a lot of importance as a proven method, due to its accuracy and sensitivity for detection of *Babesia* infection in infected host blood. Various species can be distinguished through PCR, which could not be distinguished morphologically through the blood smear method. Information about strains or subspecies may be increased through a combination of genetic sequence analysis and PCR. The required technology for individual genotyping isolates has been provided through PCR with increments in the sensitivity of parasite detection. There are six species of *Babesia* that are currently recognized in dogs [5]; three are the large piroplasms (4–5 µm) of *B. canis*, *B. vogeli*, and *B. rossi* and three are the small piroplasms (1–3 µm), *B. gibsoni, B. vulpes*, and *B. conradae.* Babesiosis is a significant tick-born disease of dogs [6]. The objective of the present study was to analyze the genetic diversity of *Babesia* species infecting selected dogs at three clinics of Lahore and one of Narowal.

## 2. Materials and Methods

Dogs of several breeds suspected for Babesiosis and presented at four different pet clinics of Lahore (clinics 1, 2, and 3 at Lahore) and Narowal (clinic 4 at Narowal) only were included in the study in the months of December, January, April, and May. Each of the animals were examined physically. A total of 200 blood samples from 117 female dogs and 83 male dogs (143 > 1 year of age and 57 < 1 year of age) were collected on a glass slide from the ear tip of the animals to prepare thin smears and these smears were dried in air. Giemsa were stained and observed under a bright field compound microscope at 1000× magnification to confirm the presence of *Babesia* species [7]. Additionally, venous blood (3 mL) was collected aseptically with anticoagulant (EDTA) from each dog. Anti-*Babesia* drugs used for treatment were recorded to follow the outcome of treatment.

### 2.1. Microscopic Detection of Babesia

Th Giemsa stained thin blood smear was observed thoroughly for the presence of *Babesia* species under a bright field compound microscope. Preliminary identification was carried out on the basis of morphological features of *Babesia* [8].

### 2.2. Polymerase Chain Reaction (PCR)

QIAamp DNA Blood Mini-Kit was used for isolation of DNA from selected blood samples of canines declared positive for Babesiosis by the microscopic method. Ultra Clean^®^ Blood DNA Isolation Kit (Non-Spin) prepared by Mo Bio Laboratories, Inc. Carlsbad, CA, USA was used for DNA extraction. Gel electrophoresis on 1% agarose gel was used to confirm the extracted DNA in each tube. DNA detection was made by using the Gel Doc™ XR+ system by Biorad USA.

Blood samples of targeted animals were processed for polymerase chain reaction to amplify the specific genes for confirmation of the protozoa. For this purpose, the procedures of DNA extraction and amplification of genes of interest were adopted as described previously [9,10,11]. Amplification of 18S rRNA of canine *Babesia* was done using primers and nested PCR protocol as described elsewhere [8]. Primer and their details for detection and differentiation among *Babesia* species, including *B. gibsoni* and *B. canis* is enlisted in (Table 1).

### 2.3. Genetic Diversity

Polymerase chain reaction was used to amplify the *B. gibsoni* (Asian genotype) and *B*. *canis* 18S rRNA genes using thermocycler (Thermo Hybaid, Middlesex, United Kingdom). A total of 25 µL reaction mixture was prepared for amplification of 18S rRNA. The reaction mixture contained 200 µM concentration of each deoxynucleotide triphosphate, 1.5 mM MgCl_2_, 0.5 µL of DNA template, 12.5 pmol of each primer, 0.625 U of Taq polymerase, and 1X concentration of PCR buffer II (Perkin Elmer) in 25 µL of each reaction. The genome was initially denatured at 95 °C for five minutes, followed by 35 amplification cycles (95 °C for one min, 65 °C for one min, and 72 °C for one min), and the step of final extension at 72 °C for five minutes. Genomic DNA of experimentally infected dogs with *B. gibsoni* (Asian genotype) and healthy dogs was used as positive and negative controls, respectively.

Under the same conditions as the outer primer pair in each separate tube, semi-nested PCR was carried out, i.e., outer reverse primer paired with specific forward primers, except for the following: As a DNA template, 0.5 µL for the starting reaction was used and for 30 s, the reactions were amplified. In separate areas, reaction, setup, sample preparation, amplicon detection, and PCR amplification were performed in order to avoid PCR amplicon contamination. In all steps, positive and negative controls were used in a 2% agarose gel containing 0.2 µg of ethidium bromide/mL. All PCR products were visualized after electrophoresis by trans-illumination using UV light. On an automated DNA sequencer i.e., LI-COR 4200 DNA sequencer by polyacrylamide gel electrophoresis (3.75%), the sequencing reactions were analyzed carefully.

### 2.4. Sequencing of PCR Products

The DNA bands corresponding to positive samples for *Babesia* spp. were excised with a fine sterile punch and the amplicons were extracted using the QIAquick Gel Extraction Kit (QIAGEN Sciences). The extracted DNA was submitted for sequencing to the Advance Bioscience International where the PCR products were sequenced on an automated DNA sequencer (LI-COR 4200 DNA), using polyacrylamide gel electrophoresis (3.75%). The sequenced chromatograms were analyzed by Chromas 2.0 and edited using BioEdit Sequence Alignment Editor, version 7.0.9.0 (Ibis Biosciences, Carlsbad, CA, USA). BLAST searches were performed in order to compare the sequences with those in the public database. A phylogenetic tree was built by mega 7 software. The evolutionary history was inferred by using the Maximum Likelihood method, based on the JTT matrix-based model.

## 3. Statistical Analysis

The data obtained in all experiments were evaluated statistically using the Statistical Package for Social Sciences (SPSS Version 20), comparing means using one-way ANOVA with Duncan’s multiple range test. The *p* value < 0.05 was considered significant.

## 4. Results

Keeping in view the importance of babesiosis and related risk factors, the present study was carried out in two districts (Lahore and Narowal). The maximum percentage of babesiosis 25/50 (50%) was recorded at clinic 4 of Narowal, followed by 22/50 (44%) at clinic 2, 20/50 (40%) at clinic 3, and 17/50 (34%) at clinic 1 of Lahore, respectively. Non-significant difference in the prevalence of babesiosis at pet clinics was observed in two districts (*p* > 0.05). Overall, the maximum percentage of positive babesiosis for clinics 1, 2, 3, and 4 was recorded in April with 34/66 (51.51%), followed by May with 25/56 (48.21%), January with 15/40 (37.50%), and December with 8/38 (21.5%). Blood samples of 84 dogs that were positive for babesiosis by microscopic examination were subjected to molecular confirmation, using genus and species-specific primers. A band of 339 bp was observed by agarose gel electrophoresis in genus-specific PCR products. Bands of 267 bp were observed in 2% agarose gel. This band indicated the positivity of genus *Babesia* causing canine babesiosis in infected dogs (Figure 1). Two *Babesia* species, including *canis* and *gibsoni*, were identified (Figure 2).

Among all the study strains, sequence similarity ranged from 93 to 100% (Figure 3). A phylogenetic tree of all the seven isolates showed significant similarity with a clear ancestral relationship on the basis of gene sequencing. Notwithstanding those phylogenic results on the basis of gene sequencing being considered the standard for classification of all the domains, there was the option to categorize the level of homology between different strains. But it was proposed that, 97–99% homology meant a similar genus and 99–100% homology means the same species [12].

The phylogenetic analysis of the seven isolates showed that MT04 and MT06 were closely related to each other, and these both had common ancestry with MT01, MT02, and MT05. On the other hand, MT08 and MT09 were also closely related to each other and were different to MT01, MT02, and MT05, but all these had common ancestry. The isolates MT01 and MT05 were closely related to the KJ15162.1 and showed 99.81% similarity, but they were not closely related to LO012799.1. Similarly, isolates MT08 and MT09 were closely related to MH100722.1 and showed less homology with MK138960.1.

## 5. Discussion

Different parasitic infections have been reported from different animals from Pakistan, and it is reported that parasitic infections pose great threat to health of animals and human [13,14,15]. Canine babesiosis is caused by intraerythrocytic parasites of the genus *Babesia*, causing a considerable worldwide threat to the canine population. It is transmitted by a tick vector, which is capable of infecting a wide variety of hosts. Even a single infected tick can be sufficient to transmit the disease [16]. The results of the present study were in accordance with previous studies [16,17,18]. Babesia was detected as 26.67% prevalence in dogs of Kerala, India [16]. Low prevalence of babesiosis (12.49 and 13.97 in 2004 and 2005, respectively) was reported in dogs presented at University Pet Center, Lahore [19]. Very low prevalence of canine babesiosis (2.62%) in 12 months of study was reported in Lahore in 1991 [20]. This variation may be due to different topographic conditions, sampling variation or detection methods.

Overall, the maximum percentage of positive babesiosis was recorded in the month of April with 34/66 (51.51%), and the results were in agreement with the studies which report higher rate of parasitic infections including *Babesia in* summer [20]. The warm and humid seasons play a key role in spreading babesiosis in dogs. Furthermore, the ticks were found active in warm seasons, spreading more babesiosis in animals [21]. The results of the study showed a close bond by reporting the highest incidence of babesiosis in canines in summer [22]. The prevalence of disease was reported as 14% through microscopy in Lahore [19]. Meanwhile, a 40% incidence of disease was reported through PCR, which is considered a highly reliable and sensitive method for molecular diagnosis [23]. The results of the study were in agreement with the results that reported 41% prevalence of babesiosis using PCR [2]. Overall, 39% infection was recorded in dogs of the present study by PCR. Figure 2 shows that our isolate sequences are closely related to the reported *B. canis* strain that was isolated from France [24] and that these strains are also similar to reported sequences of isolates reported in Poland [25]. The molecular sequence analysis with the seven isolates obtained in this study shows that they shared at least 99.2% similarity with each other. The result in this study was comparable with previous results [26]. 

## 6. Conclusions

In conclusion, only two species of *Babesia*, including *B. canis* and *B. gibsoni*, were identified in infected dogs of Lahore and Narowal, Pakistan, presented at different pet clinics.

## Figures and Tables

**Figure 1 animals-09-00439-f001:**
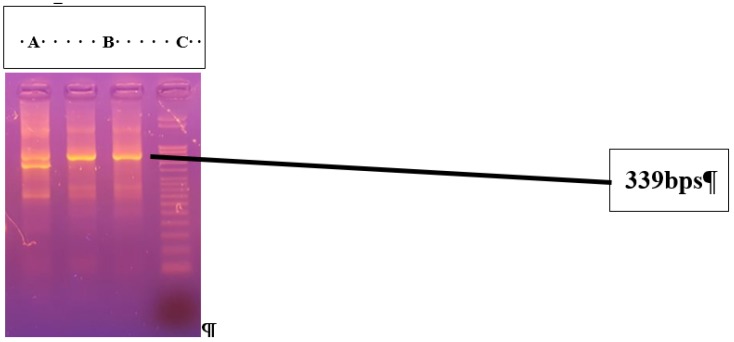
Ethidium bromide-stained agarose gel of the PCR products from genus-specific conventional PCR of *Babesia* infected dog blood samples. Lane L shows a DNA marker. Lanes from “A” to “C” are samples from naturally infected dogs showing a band of 339 bps.

**Figure 2 animals-09-00439-f002:**
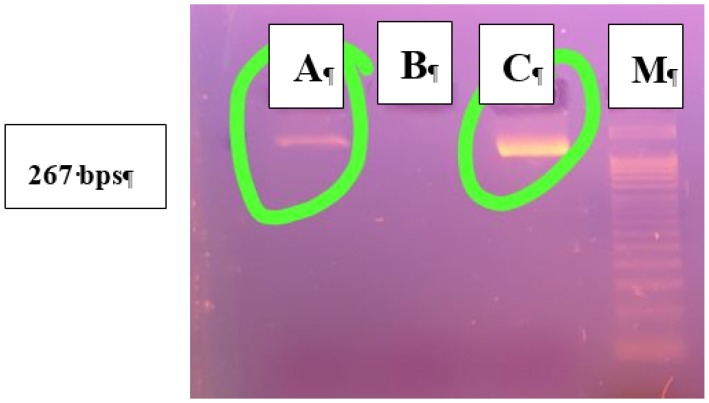
An ethidium bromide-stained agarose gel of the PCR products from species-specific nested PCR of *Babesia* infected dog blood samples. Lane M shows a DNA marker. Lanes from “A” to “C” are samples from naturally infected dogs showing a band of 267 bps.

**Figure 3 animals-09-00439-f003:**
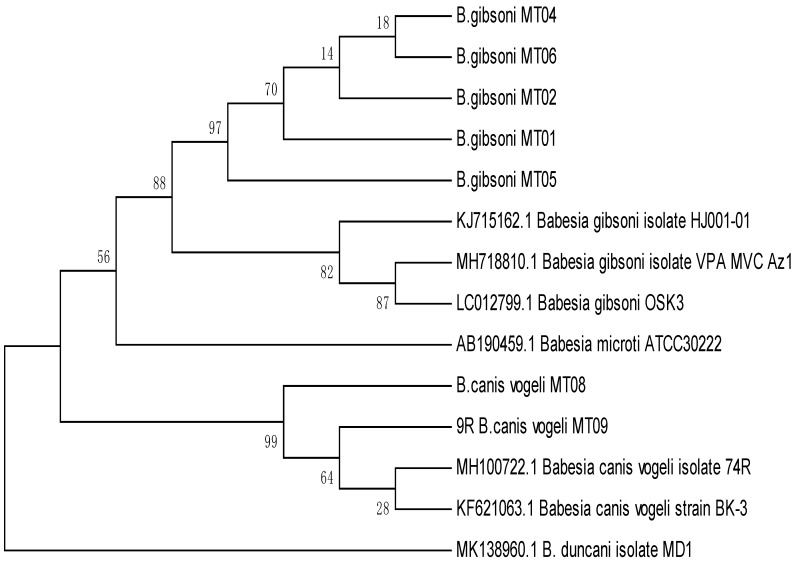
Genetic relationships of *Babesia canis* and *B. gibsoni* isolates from Pakistan with reference selected from previous studies.

**Table 1 animals-09-00439-t001:** Forward and reverse primers used in conventional and nested polymerase chain reaction (PCR).

Primer	Sequence (5′–3′)	Reaction and/or Use
5–22F	GTTGATCCTGCCAGTAGT	Full-length 18S rRNA forward primer for *Babesia* genus
1661R	AACCTTGTTACGACTTCTC	Full-length 18S rRNA reverse primer for *Babesia* genus
455–479F	GTCTTGTAATTGGAATGATGGTGAC	Semi-nested PCR outer forward primer for *B. gibsoni* Asian genotype
793–772R	ATGCCCCCACCGTTCCTATTA	Semi-nested PCR outer reverse primer for *B. gibsoni* Asian genotype
BgibAsia-F	ACTCGGCTACTTGCCTTGTC	Semi-nested PCR *B. gibsoni* (Asian genotype) specific forward primer
BCV-F	GTTCGAGTTTGCCATTCGTT	Semi-nested PCR *B. vogeli* specific forward primer
BCC-F	TGCGTTGACGGTTTGACC	Semi-nested PCR *B. canis* specific forward primer
BCR-F	GCTTGGCGGTTTGTTGC	Semi-nested PCR *B. rossi* specific forward primer
GAPDH-F	CCTTCATTGACCTCAACTACAT	Detection of PCR inhibitors
GAPDH-R	CCAAAGTTGTCATGGATGACC	Detection of PCR inhibitors

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
