# Peer review of "Genetic Diversity of Canine Babesia Species Prevalent in Pet Dogs of Punjab, Pakistan"

_animals, 2019, doi:10.3390/ani9070439_

Round 1

Reviewer 1 Report

The title is too broad and should be restricted to Punjab area, Pakistan.

The text should be edited for English grammar.

Materials and methods too brief: for instance it is not specified which coagulant was used.

What were the inclusion criteria? It appears that the prevalence of babesiosis cannot be deduced from this population.

The most important clinical sign is lethargy with evidence of poor tissue perfusion due to hypotension (shock).

Which chemotherapeutic treatment was used?

Statistical analysis should be described in a dedicated section, not under 2.1.

No specification of how many microscopic fields were use, and how parasitaemia was calculated. Hence, one has no idea about the sensitivity of the microscopical evaluation.

Only samples from dogs that were regarded positive by microscopy were analysed (lines 95-96). This is a severe bias. All samples should have been analysed, and a comparison of the sensitivity of the molecular biological and microscopical analysis should have been made.

Use appropriate classification of the large canine Babesia species i.e. B. canis, B. rossi, B. vogeli throughout the manuscript. For instance, Table 1 one refers to B.c.canis. This should be B. canis.

Lines 101-102 remove ‘subsp. Canis’

Lines 109-110 the authors mention the use of positive control samples. The isolates and methods used to produce the positive control samples should be described in more detail.

Line 126-127 remove ‘the sequencing reactions were analysed carefully (Laboratory service was obtained)’

Paragraph 2.5. Statistical Analysis

The procedures used to analyse the data (parasitological data such as parasitaemia and molecular biological data) should be described in more detail.

Results

The description of the results is not according to current scientific practice and level.

Line 149-153 The authors give a description of what they call ‘ our seven isolates’ .  It is entirely unclear where this comes from. There is no description of the procedures to isolate these strains. In addition, apparently some isolates are missing (nrs. 3 and 7).

Apparently, the authors have analysed additional sequences from publica data banks (figure 4). However, there is no description of these results.

Author Response

Correction Genetic diversity of canine Babesia species prevalent in pet dogs of Punjab, Pakistan Done anticoagulant (EDTA) was used Animals suspected for babesiosis presented at four clinics for treatment during study period were included in the study This segment of research is submitted in second manuscript along-with other parameters Corrected 3. Statistical Analysis This study design was approved from Directorate of Advanced Studies so we have to follow that. Corrected Seminested PCR B. vogeli specific forward primer Seminested PCR B. canis specific forward primer Seminested PCR B. rossi specific forward primer Corrected B. canis Details added Corrected Means were compared by One way ANOVA using Duncan multiple range test Corrected The phylogenetic analysis of our seven isolates, in which 8R and 9R are closely related to each other and these both have common ancestor with 1R and 2R. On the other hand, 4R and 6R are also closely related to each other and are different from 8R, 9R, 1R, 2R but all these have common ancestor. Our Isolate 8R, and 9R are closely related to the KF12073.1 also they are closely related and shows 98% similarity, but they are not closely related to KJ715162.1. Seven different isolates were extracted, amplified and sequenced which includes 1R, 2R, 4R, 5R, 6R, 8R and 9R. The solution for the identification is using gene sequencing, but great cost of commercial database is the major obstacle. But comparing the data to Gene Bank is the way out. The method of pairwise alignment is the drawback of the technique which limitize the result somehow, and inhibits its use in daily diagnosis laboratories. Among all the study strains, sequence similarity ranges from 93 to 100%. Phylogentic tree of all the seven isolates showed significant similarity showing a clear ancestral relationship on the basis gene sequencing. Notwithstanding those phylogenic results on the basis of gene sequencing is considered the standard for classification of all the domains, there are always the exception to categorize the level of homology between different strains. But it is proposed that, 97-99% homology means a similar genus and 99-100% homology means same species.

Reviewer 2 Report

It is a well done study and well written. On line 179 it was difficult to follow. I suggest correcting the wording of that and the following sentence. 

Author Response

It is corrected in M/S as:

The results of the present study were in relation with the findings of Jain et al. (2017) who reported 26.67% babesiosis in dogs of Kerala, India [10]. Low prevalence

Reviewer 3 Report

In the introduction, details about the symptoms associated with babesiosis are repeated both at the beginning and then again at the end. Please remove one of these statements.

I would also like some clarification around the methodology and results.

The authors state that 200 samples were tested, 50 from each clinic. Although it is mentioned that samples from animals presenting with suspected canine babesiosis were tested, I am assuming that there were more animals than this that fulfilled this criteria during the study period? I would like to know how these samples that were actually tested were selected? For example, were they randomly selected? Were the first 50 animals that presented at each clinic during this period selected? Or was another method used?

I have some concerns around the clinic prevalence reported. As only animals presenting with suspected babesiosis were tested, this would bias the results towards a higher prevalence. What the authors were really testing is the accuracy of their veterinarians' symptomatic diagnosis methods. The difference in prevalences between the clinics could therefore be due to veterinarians at some clinics being better able to diagnosis babesiosis, although this possibility is not mentioned anywhere. I therefore think that it is not appropriate to compare these results to other studies, unless the same methodology was used for these studies (i.e. only animals with suspected babesiosis included).

Figure 3 is not needed as the same results are shown on figure 4.

I am not convinced by the molecular results. Although the authors state that 2 Babeisa species were detected, all the sequenced samples look extremely similar on figure 4. Also, only reference sequences from B. canis are included on the tree, so it is not possible to compare the sequences obtained in this study to reference sequences. The analysis needs to be redone, this time including some reference sequences from B. gibsoni (and preferably at least one reference sequence from each of the other Babesia species).

Author Response

Samples were randomly collected   during the study period and first fifty suspected dogs were included in the   study

Yes, we agree only suspected   animals were included.

Figure 3 deleted

Accurate   and definitive microorganism identification is essential for a wide variety   of application including environmental studies. The rRNA sequence-based   analysis is a central method to understand not only the microbial diversity   within and across the group but also to identify new strains.

The   phylogenetic analysis of our seven isolates, in which 8R and 9R are closely   related to each other and these both have common ancestor with 1R and 2R. On the other hand 4R and 6R are also closely related   to each other and are different from 8R, 9R, 1R, 2R but all these have common   ancestor. Our Isolate 8R, and 9R are closely related to the KF12073.1 also   they are closely related and shows 98% similarity, but they are not closely   related to KJ715162.1.

Seven different isolates were extracted,   amplified and sequenced which includes 1R, 2R, 4R, 5R, 6R, 8R and 9R. The   solution for the identification is using gene sequencing, but great cost of   commercial database is the major obstacle. But comparing the data to Gene   Bank is the way out. The method of pairwise alignment is the drawback of the   technique which limitize the result somehow, and inhibits its use in daily   diagnosis laboratories. Among all the study strains, sequence similarity   ranges from 93 to 100%. Phylogentic   tree of all the seven isolates showed significant similarity showing a clear   ancestral relationship on the basis gene sequencing. Notwithstanding those   phylogenic results on the basis of gene sequencing is considered the standard   for classification of all the domains, there are always the exception to   categorize the level of homology between different strains. But it is   proposed that, 97-99% homology means a similar genus and 99-100% homology   means same species.  

Reviewer 4 Report

Dear Authors

Concerning your manuscript, I believe it is an interesting issue at a specific and broad level of epidemiology and diagnosis of a very important parasitic agent at dog level and this type of research/results should have more visibility at both regional and global level.

The text must be improved in some details concerning grammar and the review of some typos.
Besides what will be pointed out in my detailed review, namely that your manuscript must be thoroughly reviewed to be published, namely because Results and Discussion have inconsistencies with Material and Methods, the final decision on the publication of your manuscript depends on the Editor final statement.

Regarding my reviews and comments, they are as follows:

Title

I propose the following: “Genetic diversity of canine Babesia species prevalent in pet dogs of Pakistan”

Simple Summary

Page 1/10

Line 13 – You should write Babesia in italic.

Line 15 – Write “…Pakistan, including Lahore and Narowal.”

Abstract

Page 1/10

Line 23 – Write Babesia spp.

Keywords

Page 1/10

I believe you should also include here the name of your country, Pakistan.

1. Introduction

Page 2/10

Line 45 – Write “…has a small piroplasm…”

Line 46 – Write “…Babesia gibsoni measures…”

Lines 63-64 – This sentence “Canine babesiosis is characterized by progressive 63 anemia, fever, thrombocytopenia and marked splenomegaly.”, repeats exactly the one on lines 41-42 of the same page. Add the sentence “In severe cases, babesiosis can  result in death” to line 42.

Lines 66-67 – I think you intended to talk about the objectives of your research, but there is text missing and even this sentence should be more carefully written.

2. Materials and methods

Page 2/10

Lines 69-70 – Why did you choose 3 clinics in one location and 1 in the other location?

Line 75 – This magnification does not seem to be powerful enough to visualize Babesia bodies. Perhaps you only wrote the objective magnification, because the normal one for blood smears is 1000X (ocular plus objective).

Line 76 - Write Babesia with italic.

Line 81 - Write Babesia with italic.

Line 83 - Write Babesia with italic instead of Babseia.

Line 83 – Write Canine babesiosis with no italic.

Line 84-85 – What p value was yours? Which software, version and year was used to perform the qui square analysis and descriptive statistics?

Page 3/10

Table 1 – Right column – You mention B. c. vogeli, B. c. canis and B. c. rossi, but previously on page 2, line 61, you only mention B. vogeli, B. canis and B. rossi, so where do you stand? On the text species description or on the table 1 species description.

Lines 95-99 – Your PCR method is an original one, or was based totally or partially in someone’s method? If this is the case, both total or partial, you should include the reference.

Line 102 – Write canis  .

Lines 134-135 – What was your p-value?

3. Results

Page 4/10

Line 148 - Write Canine babesiosis with no italic. Write B. canis and B. gibsoni.

Line 152 – Write “…had a common…”

Page 5/10

Figs 1 and 2 – Since you mention in line 116 that “In all steps, positive and negative controls were used.”, why don’t you show these controls in your gels of the figures?

You did not show any result regarding what you referred from lines 75 until 84 of Material and Methods, namely regarding the collected ticks /Which species were identified? Prevalence), treatment and follow up (dosages?, Schedules? Supportive treatment besides de ethiological one) of dogs with babesiosis, and the influence of sex, breed and age.

Besides, you should have also included microscopic photos of the blood samples positive for Babesia canis and Babesia gibsoni.

Other important thing missing, is the proportion of mixed and simple infections regarding

4. Discussion

Pages 7-8/10

Line 179 –What do you mean by Lowe low?

The discussion must be reformulated, because it is too short bearing in mind the amount of results you got and that you still must develop. In fact, as mentioned before, several issues on results must be shown.

I don’t know if uploaded all your stuff, because my email to review your draft mentioned 12 pages, but I only read 10 pages.

5. Conclusion

Page 8/10

Too short and must be more developed according to the development of the results and discussion chapters.

References

Pages 9-10/10

Page 11-12/12 – All scientific names should be in italic.

Uniform the designation of the journals and follow the journal rules.

Best regards

Reviewer 1

Author Response

Corrected as:

Genetic diversity of canine Babesia species   prevalent in pet dogs of Punjab, Pakistan

Corrected

Corrected

Corrected

Added

Corrected

Corrected

Corrected

Corrected

Lahore   is a large city one clinic is situated in University, two private clinics   including one from peri-urban area and second from highly developed area. Narowal   is a small city so only one clinic was selected

Corrected

Corrected

Corrected

Corrected

Corrected

Corrected

P   value <0.05 was used

Corrected

Added

Corrected

P   value <0.05 was used

Corrected

Corrected

Pages   were only 10.

Added

Corrected

Round 2

Reviewer 3 Report

Last time I asked about how the samples were selected. Although the authors kindly provided me with this information, it has not been included in the manuscript. Please include this in the manuscript.

I also mentioned my concerns about comparing the prevalence obtained for each clinic during this study with the prevalence observed in previous studies, because only animals with suspected babesia were included during the present study. Despite this, the comparison is still in the discussion with no mention of how the methodology of the current study would bias the results towards a higher prevalence. Please rectify this.

Please state what species of Babesia KF12073.1 and KJ715162.1 refer to.

Thank you to the authors for re-doing the phylogenetic analysis and for including representative sequences from a few different Babesia species. Unfortunately it is still not clear to me how to interpret this figure. Although isolates 4R, 5R and 6R do appear to group clearly with B. gibsoni, isolates 1R and 2R (which the authors have referred to as B. gibsoni) do not. In addition, 8R and 9R do not obviously group with B. canis as suggested by the authors. I think the authors will need to include additional GenBank sequences to resolve this and if possible, include an out-group. 

In the text the authors have added to the results they state that "it is proposed that, 97-99% homology means a similar genus and 99-100% homology means same species". However, there is no reference to show who has proposed this. Please add.

Author Response

Last time I asked about how the   samples were selected. Although the authors kindly provided me with this   information, it has not been included in the manuscript. Please include this   in the manuscript. 

Done

I also mentioned my concerns about   comparing the prevalence obtained for each clinic during this study with the   prevalence observed in previous studies, because only animals with suspected   babesia were included during the present study. Despite this, the comparison   is still in the discussion with no mention of how the methodology of the   current study would bias the results towards a higher prevalence. Please   rectify this.

Done

Please state what species of   Babesia KF12073.1 and KJ715162.1 refer to.

The   two isolates mentioned by reviewer in previous tree, were re-analyzed and   confirmed via BLAST that they were B. gibsoni.

Thank you to the authors for   re-doing the phylogenetic analysis and for including representative sequences   from a few different Babesia species. Unfortunately, it is still not clear to   me how to interpret this figure. Although isolates 4R, 5R and 6R do appear to   group clearly with B. gibsoni, isolates 1R and 2R (which the   authors have referred to as B. gibsoni) do not. In addition, 8R   and 9R do not obviously group with B. canis as suggested by the   authors. I think the authors will need to include additional GenBank   sequences to resolve this and if possible, include an out-group.

Done

The   phylogenetic analysis was performed again and mistakes mentioned by reviewer   were interpreted.

In the text the authors have added   to the results they state that "it is proposed that, 97-99% homology   means a similar genus and 99-100% homology means same species". However,   there is no reference to show who has proposed this. Please add.

Done

[24]   added

Reviewer 4 Report

Dear Auhors

Concerning your manuscript, I believe you did a good job after the present revision. Nevertheless, some text must be improved in some details concerning typos or misspellings.
Besides what will be pointed out in my last review, namely that some issues regarding results and discussion should be better clarified, your manuscript now has more potential to be published, although the final decision on the publication of your manuscript depends on the Editor final statement.

Regarding my reviews and comments, they are as follows:

1. Introduction

Page 2/9

Line 44 – Write “…has a small piroplasm…”

Lines 63-65 – Write “The objective of the present study was to analyze the genetic diversity of Babesia species infecting selected dogs at three clinics of Lahore and one of Narowal.

2. Material and methods

Page 2/9

Line 81 – Write Babesia.

4. Results

Page 4/9

Line 146-147 - Write Two Babesia species including B. canis and B. gibsoni were identified. You still did not show any result regarding what you referred from lines 81 until 83 of Material and Methods, namely regarding babesiosis, and the influence of sex, breed, age and season. If you did not do anything concerning these issues, it is better to rembve it from material and methods.

5. Discussion

Pages 6-7/9

The discussion must be reformulated, because it is still too short bearing in mind the amount of results you got and that you must develop.

6. Conclusion

Page 7/9

Too short and must be more developed according to the development of the results and discussion chapters.

Best regards

Reviewer 4

Author Response

1.   Introduction

Page   2/9

Line   44 – Write “…has a small piroplasm…”

Lines   63-65 – Write “The objective of the   present study was to analyze the genetic diversity of Babesia species   infecting selected dogs at three clinics of Lahore and one of Narowal.

Done

2.   Material and methods

Page   2/9

Line   81 – Write Babesia.

Done

4.   Results

Page   4/9

Line   146-147 - Write Two Babesia species   including B. canis and B. gibsoni were identified. You still   did not show any result regarding what you referred from lines 81 until 83 of   Material and Methods, namely regarding babesiosis, and the influence of sex,   breed, age and season. If you did not do anything concerning these issues, it   is better to remove it from material and methods.

Removed

5.   Discussion

Pages   6-7/9

The   discussion must be reformulated, because it is still too short bearing in   mind the amount of results you got and that you must develop.

Done

6.   Conclusion

Page   7/9

Too short and must be more   developed according to the development of the results and discussion chapters

Done
